# Functions of ROS in Macrophages and Antimicrobial Immunity

**DOI:** 10.3390/antiox10020313

**Published:** 2021-02-19

**Authors:** Marc Herb, Michael Schramm

**Affiliations:** Faculty of Medicine, University Hospital Cologne, Institute for Medical Microbiology, Immunology and Hygiene, University of Cologne, 50935 Cologne, Germany; marc.herb@uk-koeln.de

**Keywords:** macrophages, reactive oxygen species, mitochondria, NADPH oxidases, infection, redox signaling, antimicrobial defense, inflammasome, immunity, ROS scavenging, ROS detection

## Abstract

Reactive oxygen species (ROS) are a chemically defined group of reactive molecules derived from molecular oxygen. ROS are involved in a plethora of processes in cells in all domains of life, ranging from bacteria, plants and animals, including humans. The importance of ROS for macrophage-mediated immunity is unquestioned. Their functions comprise direct antimicrobial activity against bacteria and parasites as well as redox-regulation of immune signaling and induction of inflammasome activation. However, only a few studies have performed in-depth ROS analyses and even fewer have identified the precise redox-regulated target molecules. In this review, we will give a brief introduction to ROS and their sources in macrophages, summarize the versatile roles of ROS in direct and indirect antimicrobial immune defense, and provide an overview of commonly used ROS probes, scavengers and inhibitors.

## 1. Reactive Oxygen Species

The term reactive oxygen species (ROS) describes a group of molecules with at least one oxygen atom and with higher reactivity than molecular oxygen (O_2_). This group consists of two subclasses: (I) highly reactive free radicals, including the superoxide anion (O_2_^●−^), the hydroxyl radical (●OH), alkoxyl (●OOR) and peroxyl radicals (●OOH) [1,2,3] and (II) nonradical species such as hydrogen peroxide (H_2_O_2_), singlet oxygen (^1^O_2_) [4,5], ozone (O_3_), and the hypochlorite anion (OCl^−^) [6,7]. ROS are produced by nearly all organisms and cells [8,9,10,11,12]. O_2_^●−^, the common precursor of all ROS produced by cells, is produced by single electron transfer to O_2_. O_2_^●−^ quickly dismutates to H_2_O_2_, either spontaneously in the presence of water or catalyzed by superoxide dismutases [13,14] (Figure 1).

O_2_^−^ and H_2_O_2_ are the two most abundant ROS subspecies in cells, but highly differ in their chemical parameters and, therefore, in their behavior and function.

O_2_^−^ shows higher reactivity than H_2_O_2_ and cannot cross cell membranes, except through ion channels, such as voltage-dependent anion channels (VDAC) [15]. An increase in O_2_^●−^ levels is associated with oxidative stress and cellular damage [16,17,18,19], such as oxidation of proteins [20,21,22], amino acids [23] and DNA [19,24] or lipid peroxidation [25,26,27]. Moreover, O_2_^●−^ can irreversibly inactivate proteins and thereby contribute to cellular signaling [28].

H_2_O_2_, as a diffusible and relatively stable molecule, is more suitable as a cellular signaling factor than O_2_^●−^ [29,30,31,32,33]. Although H_2_O_2_ is more diffusible than O_2_^●−^, its diffusion over membranes is limited [34,35]. This questions the view of saturation of the cell with H_2_O_2_ to induce signaling pathways without regarding the compartment of ROS production [36,37]. Aquaporins allow a faster and controlled passage of H_2_O_2_ over membranes and add another regulatory level to ROS-mediated signaling [38,39]. The majority of H_2_O_2_-mediated signaling is based on the oxidation of cysteine residues [30,40,41,42,43,44,45]. At physiological pH, thiol groups of cysteines (Cys-SH) exposed to the cytosol are deprotonated to thiolate groups (Cys-S^−^), which are susceptible to oxidation in dependency of their pKa [46,47]. H_2_O_2_-dependent signaling occurs at nanomolar concentrations (≈100 nM), leading to reversible oxidation of the thiolate group to a sulfenic group (Cys-SOH). Protein oxidation by H_2_O_2_ can lead to allosteric changes that alter binding affinity for substrates or promote or inhibit enzymatic function [48,49,50,51]. Moreover, it can lead to covalent linkage of cysteine residues by disulfide bonds (Cys-S-S-Cys) [30,40,52]. Such protein oxidations can be reversed by the antioxidant defense system and therefore function as important redox switches in various cellular processes [53,54]. Excessive H_2_O_2_ production, however, leads to further oxidation of the sulfenic group (Cys-SOH) to sulfinic (Cys-SO_2_H) and sulfonic groups (Cys-SO_3_H), a process that is irreversible and results in protein malfunction [55].

## 2. Cellular Redox Balance

In cells, ROS levels are in a dynamic but stable equilibrium that consists of the sum of all producing and eliminating factors. ROS are produced as byproducts of oxidative metabolic processes [56,57] and in response to exogenous signals such as pathogens [30,58,59,60,61,62] or endogenous signals such as cytokines [63,64].

The cellular antioxidant defense system balances ROS production. The most prominent components are the three superoxide dismutases SOD1, SOD2 and SOD3, which catalyze the dismutation of O_2_^●−^ to H_2_O_2_ and catalase, which detoxifies H_2_O_2_ to H_2_O and O_2_ [11,65,66,67]. While catalase is present in peroxisomes, cytosol and mitochondria [65], the SODs show strict subtype-dependent compartmentalization. SOD1 localizes to the cytosol and the intermembrane space of mitochondria (IMS), SOD2 is present exclusively in the mitochondrial matrix and SOD3 is exclusively extracellular [13,68]. Various soluble factors, such as glutathione (GSH) (in combination with the glutathione oxidase) and thioredoxin (in combination with thioredoxin reductase) [69,70,71,72,73], as well as membrane-integrated molecules, such as the vitamin E α-tocopherol (see Section 6) and coenzyme Q (CoQ), contribute to cellular antioxidant defense by scavenging ROS [41,66,74].

ROS levels that exceed the capacity of the cellular antioxidant defense system induce oxidative stress [29,75]. Excessive oxidative stress causes damage to cellular components such as proteins, DNA and lipids and results in a number of different pathologies [76,77,78]. In this case, it is also referred to as oxidative distress (Figure 2). However, oxidative stress is not detrimental per se. Generation of oxidative stress in pathogen-containing phagosomes of phagocytes, for example, is a crucial component of antimicrobial immunity [58,61,79]. Moreover, growing evidence suggests that a tightly controlled increase in cellular ROS levels can actually have beneficial consequences for the cell, a condition that has been termed oxidative eustress [29,75]. Oxidative eustress has important signaling functions in several immunological, physiological and cellular processes [30,46,80,81,82] (see also Section 4.1, Section 4.2 and Section 4.3). Thus, maintaining the delicate balance between oxidative eustress and distress is essential for cell function [80] (Figure 2).

## 3. Cellular ROS Sources

The enzyme family of NADPH oxidases (Nox) [13,83] and the respiratory electron transport chain (ETC) of mitochondria [84,85,86] are the two major sources of ROS in many eukaryotic cells.

### 3.1. NADPH Oxidases

The Nox enzyme family consists of the isoforms Nox1, Nox2, Nox3, Nox4, Nox5, and the dual oxidases Duox1 and Duox2. All isoforms are integral membrane proteins and transfer electrons from NADPH to O_2_, thereby generating O_2_^●−^. In fact, generation of ROS is the sole purpose of all Nox isoforms. Nox1, Nox2, Nox3, and Nox4 depend on a common subunit, p22^phox^ [87,88,89]. Nox5, Duox1 and Duox2 are independent of p22^phox^. They are directly activated through calcium binding to their EF-hand calcium-binding domain [87,90,91] (Figure 3).

Nox enzymes have various functions in many organisms from organ to cellular level including antimicrobial defense, vasoregulation, hormone synthesis, and regulation of gene expression, cell proliferation and differentiation [13,83,92]. In particular Nox2 can produce large quantities of ROS, which are a crucial component of the antimicrobial activity of professional phagocytes, including macrophages [58,93]. Nox1, 3, 4, 5, and Duox1 and 2 usually produce smaller amounts of ROS for regulation of signaling pathways or during anabolic processes such as endothelial vasoregulation or synthesis of thyroid hormones [94,95,96,97,98]. As an exception, we could recently show that Nox1 is also a source for large quantities of ROS in activated proinflammatory microglia [99].

### 3.2. Mitochondria

Mitochondria produce ROS during respiratory activity [13,57]. Electrons are shuttled from NADH and FADH_2_ through the four complexes I-IV of the ETC, generating a proton gradient that drives ATP production. Mitochondria-derived ROS (mtROS) are mainly produced by complexes I and III as O_2_^●−^. Complex I produces O_2_^●−^ into the mitochondrial matrix where it is dismutated by SOD2 to H_2_O_2_ [67,84,85,86]. We will refer to these ROS as matrix mtROS (Figure 4).

Complex II is also capable of matrix mtROS production [100,101,102,103,104] and two different modes of production were described: (I) complex II delivers electrons to other ETC complexes, where mtROS production takes place, for example during reverse electron flow (RET) to complex I, where vast quantities of matrix mtROS are formed [105,106,107]. This complex II-fueled mtROS production at complex I is still controversial and could be tissue-specific [108]. (II) Complex II directly produces matrix mtROS at its matrix-located FAD binding site [109,110,111], but the exact mechanism is not fully investigated yet. Both indirect and direct matrix mtROS production by complex II was discovered several years ago, but remained mainly unregarded [86,112,113]. After mutations of complex II were identified that resulted in increased matrix mtROS production with critical consequences during cancer and several diseases [114,115,116,117] its investigation switched more into focus [103,118,119].

Complex III, as exception to complexes I and II, not only produces O_2_^●−^ into the matrix but also directly into the intermembrane space (IMS) [86,120]. From there it reaches the cytosol via VDAC [15,121] or it is dismutated by IMS-located SOD1 to H_2_O_2_, which reaches the cytosol via diffusion or aquaporins [85,122,123,124,125,126]. Importantly, due to the high efficiency of the matrix-located antioxidant defense system, matrix mtROS cannot escape from intact mitochondria, which was confirmed for isolated mitochondria as well as in cells [127,128,129]. Membrane-disrupting damage of mitochondria or opening of the mitochondrial permeability transition pore (mPTP) is necessary for matrix mtROS release into the cytosol [127,128,129,130,131]. We will refer to both, direct mtROS production into the cytosol via complex III and matrix mtROS that are clearly released from mitochondria, as cytosolic mtROS (Figure 4).

For a long time, production of mtROS was described as an unavoidable byproduct of respiratory activity and increased mtROS levels viewed only as causing oxidative distress [132,133,134,135]. In recent years, however, several studies have shown that mtROS production is actively triggered by a number of different stimuli including the recognition of pathogens and pathogen-associated molecular patterns (PAMPs) [30,61] or endogenous molecules such as TNF [48,136].

### 3.3. Other ROS Sources

This review focuses on mitochondria and Nox enzymes as the most prominent and best characterized ROS sources in macrophages during antimicrobial responses. However, also other ROS sources are present in macrophages, which are discussed in this section.

#### 3.3.1. Xanthine Oxidase

Xanthine oxidase (XO) is a well characterized enzyme [137,138], mainly located in the cytosol, which fulfills a major role in purine nucleotide catabolism [139]. It catalyzes the oxidation of the bases hypoxanthine and xanthine to uric acid, which is the end product of purine catabolism. H_2_O_2_ is generated as a byproduct during two oxidation steps of the enzymatic reaction, marking XO as a potential ROS source. XO has important physiological functions [140,141,142,143], however, only a few studies investigated its role in macrophages and suggested involvement in inflammasome activation [144], regulation of cytokine expression [145] and defense against parasitic infection [146].

#### 3.3.2. Peroxisomes

Peroxisomes are cytosolic organelles present in nearly all eukaryotic cells. Due to their oxidative metabolism of fatty acids and amino acids, they are in close contact and communicate with mitochondria [147,148,149]. During the β-oxidation of fatty acids, O_2_^●−^ and, subsequently, H_2_O_2_ are generated as byproducts [147]. With high concentrations of catalase, glutathione oxidase and SOD, peroxisomes do not only detoxify ROS produced by themselves, but also function as a ROS sink for the cytosol [150,151,152] suggesting a role for peroxisomes in regulation of the redox balance of the cell and therefore in redox-dependent signaling [148,153]. Moreover, peroxisomes are necessary for the degradation of hormones such as leukotrienes [154,155] and prostaglandins [156], which both have immunoregulatory functions [157,158]. Peroxisomes also have potential roles in resolution of inflammation, since they generate polyunsaturated fatty acids that serve as precursor molecules for anti-inflammatory substance classes such as resolvins, maresins and protectins [159,160,161]. The important role of peroxisomes in immunity was investigated and brought to mind by many studies [162,163,164]. However, so far, only one study investigated involvement of peroxisome-dependent ROS production in macrophage-mediated immunity and suggested a role in direct antibacterial defense and phagocytosis in the fruit fly *Drosophila melanogaster* [165].

#### 3.3.3. Cyclooxygenases and Lipoxygenases

The arachidonic acid (AA)-metabolizing enzyme families of cyclooxygenases (COX) and lipoxygenases (LOX) both generate ROS as a byproduct. The two COX isoforms, COX1 and COX2, are membrane-bound enzymes that catalyze the conversion of AA in a bifunctional reaction [166]. In the first step, a cyclic peroxide bond is incorporated by the cyclooxygenase activity to release prostaglandine-G2 as intermediate. O_2_ is consumed as cosubstrate and H_2_O_2_ is generated. In the second step, prostaglandine-G2 is reduced to prostaglandine-H2 by peroxidase activity and the previously generated H_2_O_2_ is used as cosubstrate. COX enzymes are therefore crucial for the biosynthesis of prostaglandins, which exert many physiologic functions, such as vasodilation, generation of eicosanoid hormones, thermoregulation, digestive regulation, and induction of fever [167,168,169]. COX1 is constitutively expressed in many tissues [170,171], whereas COX2 is an inducible enzyme with low or undetectable tissue levels under homoeostatic conditions [172,173]. Nevertheless, connections of COX2 to immunity were suggested because macrophages can produce huge amounts of prostaglandins [174,175,176,177,178]. Moreover, COX2 expression in macrophages can be induced by various proinflammatory stimuli such as lipopolysaccharide (LPS), interleukin-1β (IL-1β), interferon γ (IFNγ), NO, and pharmacologically by phorbol-12-myristat-13-acetat (PMA) [158,176,179,180,181,182] as well as during parasitic infection [183,184]. However, while listed as potential ROS sources in macrophages, no role for COX-dependent ROS production during immune responses has been identified so far. On the contrary, ROS produced by other sources were discussed as inducers of COX enzyme expression and activity. Nox2-derived ROS [185] and matrix mtROS [186] were suggested to induce COX. However, this suggestion was based exclusively on data obtained with diphenyl iodinium (DPI), a general flavoprotein inhibitor with various side effects and with lack of specificity (see Section 7). Therefore, the findings are not fully conclusive. So far, the COX-inducing ROS sources still remain elusive. The family of LOX enzymes comprises various isoforms across plants and animals [187,188]. In mammals, LOX catalyze the production of leukotrienes from AA with ROS as byproduct [189,190]. The LOX isoform 5-lipoxygenase is the best characterized LOX in mammals and suggested as a main source of intracellular ROS production besides mitochondria and Nox enzymes [191,192,193]. However, a role for LOX-mediated ROS production in macrophages remains to be demonstrated.

#### 3.3.4. Cytochrome P450 Enzymes

Cytochrome P450 (CYP) enzymes are heme monooxygenases [194] and named after the absorption band at 450 nm observed after binding of carbon monoxide [195,196,197]. They are found across all members of the three life domains, bacteria, archea and eukarya [198,199,200,201,202] and have crucial roles in detoxification of drugs and xenobiotics but also in the anabolism of sterols, fatty acids, eicosanoids, and vitamins [203,204]. CYP enzymes have also been shown to generate, similar to COX and LOX enzymes, prostaglandins and leukotrienes from AA as substrate as well as products from concerted reactions with LOX, like hydroxyeicosatetraenoic acids [205,206]. CYP enzymes can catalyze a wide range of reactions but the most common reaction is the oxidation of a substrate, which consumes O_2_ and a variable electron donor. In the active center of CYP enzymes, several radical molecule species can be generated as byproducts, including ROS [207,208]. ROS generation particularly occurs via two shunts within the CYP catalytic cycle, during which the oxidation reaction of the substrate is not finished and no product is generated [209]. The first shunt releases O_2_^●−^, which then quickly dismutates to H_2_O_2_ [210]. During the second shunt H_2_O_2_ is directly formed through reduction of O_2_^●−^ by addition of one H atom [210]. While extensively studied and excellently reviewed [211], so far CYP enzymes were mainly described as inducers of oxidative distress, leading to various diseases such as lung injury [212,213] and hepatotoxicity [214,215]. Expression of various CYP enzymes in macrophages was shown over 20 years ago [216], however, reports of roles for CYP enzymes in inflammatory responses of macrophages are rare [217,218] and address the regulatory intermediates generated by CYP rather than the role of CYP-mediated ROS generation. Surprisingly, nothing is known about this topic so far.

## 4. Macrophages and ROS

Macrophages often are the first immune cells that encounter invading pathogens [219,220,221]. They engulf pathogens, dead cells and cellular debris by phagocytosis and subsequently degrade the cargo in phagolysosomes [222,223,224]. Beyond the production of ROS, macrophages also employ an array of directly antimicrobial mechanisms, e.g., the generation of reactive nitrogen species (RNS) in the phagosome and the delivery of cathepsins and other hydrolases into maturing phagosomes [59,225,226,227,228,229,230]. Indirect antimicrobial mechanisms include the activation of inflammasomes and the secretion of cytokines and chemokines, which help to orchestrate the subsequent innate and adaptive immune responses [30,231], and the MHC-dependent presentation of pathogen-derived antigens [232]. In this review, we will focus on the different ways of how macrophages use ROS for antimicrobial defense.

### 4.1. Direct Antimicrobial Functions of ROS in Macrophages

#### 4.1.1. ROS vs. Bacteria

Recognition of bacteria by macrophages leads to ROS production in different cellular compartments, where they fulfill different antibacterial functions. One of the first functional roles described for ROS produced by macrophages was the inactivation of phagocytosed bacteria by the oxidative burst generated by Nox2. Recognition of invading bacteria induces a fast and robust production of ROS into the extracellular space and the phagosomal lumen (extracellular ROS) (see Section 5.2) [30,58,233,234,235]. This is completely abrogated in Nox2-deficient peritoneal macrophages (PM) and bone marrow-derived macrophages (BMDM) [30,58,232] establishing Nox2 as the exclusive source of extracellular ROS produced by macrophages in response to bacterial infection. Individuals with chronic granulomatous disease (CGD), a genetic disorder caused by hypo- or amorphic mutations in genes encoding for Nox2 or its subunits, fail to produce sufficient amounts of ROS by Nox2 and in consequence suffer from increased susceptibility to infections [236] underscoring the importance of Nox2-derived ROS for antimicrobial immunity. In dependence of the invading pathogen, also other Nox isoforms or mitochondria can be activated to produce ROS either directly into the phagosome to inactivate phagocytosed bacteria or ROS are produced into the cytosol to counter bacteria, which already escaped from the phagosome.

*Listeria monocytogenes* (L.m.), a food-borne pathogen, specializes in escaping from the phagosome via its pore-forming toxin listeriolysin O and two phospholipases PlcA and PlcB [237,238,239]. During infection with L.m., tissue macrophages activate a highly antimicrobial phagocytic pathway, LC3-associated phagocytosis (LAP), during which phagosomes become decorated with LC3 [240,241,242]. These so-called LAPosomes show enhanced fusion with lysosomes, leading to improved killing of L.m. and in consequence substantially improve immunity to L.m. in vitro and in vivo (see Figure 5) [58,243]. LAP activation strictly requires Nox2-derived extracellular ROS production, but notably, Nox2-derived ROS do not seem to be listericidal by themselves [58]. ROS production by Nox2 during listerial infection is regulated on different levels. Receptor-mediated recognition of the invading bacterium is often the first step in the cascade that initiates the antibacterial capacities of macrophages. For L.m., the integrin Mac-1 is crucial for initiation of ROS production by Nox2, whereas Toll-like receptor (TLR)-mediated signaling is completely dispensable [58]. On the cytosolic level, deubiquitinases (DUBs) have been shown to negatively influence Nox2-mediated ROS production during L.m. infection in the immortalized macrophage-like cell line RAW246.7 (RAW) [244]. Chemical inhibition of DUBs in RAW cells infected with L.m. led to an increase in total cellular ROS levels (see Section 5.1) and reduced bacterial burden. Nox2-deficient RAW cells showed no induction of total cellular ROS production after DUB inhibition and no reduction in bacterial burden indicating that DUBs restrict Nox2-dependent ROS production and bacterial clearance. The negative regulator of ROS (NRROS), a regulatory factor directly interacting with Nox2, negatively influences ROS production via Nox2 by competing with p22^phox^ in binding gp91^phox^ [245]. Priming with LPS, IFNγ or tumor necrosis factor (TNF) resulted in downregulation of NRROS and therefore enhanced production of total cellular ROS in BMDM coincubated with heat-killed L.m. Accordingly, NRROS-deficient BMDM showed increased total cellular ROS levels after coincubation with heat-killed L.m. Not only Nox2 but also mitochondria can be recruited as source for phagosomal ROS during bacterial infection [61]. Geng and colleagues showed that this is also the case during L.m. infection [246]. The study identified two phagosome-located kinases, mammalian sterile 20-like kinases 1/2 (MST1/MST2), which regulate mitochondrial recruitment to L.m.-containing phagosomes. MST1/2-deficient BMDM showed reduced ROS production into phagosomes and increased bacterial burden. *Escherichia coli* (E.c.) coupled to the ROS probe CellROX (see Section 5.1) were used to measure mitochondrial contribution to phagosomal ROS levels. Notably, MST1/2 did not directly regulate mtROS production since matrix mtROS levels remained unaltered, as determined with MitoSOX (see Section 5.2). These studies show that ROS production either from Nox2 or from mitochondria are crucial players for antilisterial defense (Figure 5).

In contrast to the clear role of Nox2-dervied ROS in antilisterial activity of macrophages, the role of Nox2 during infection with another Gram-positive pathogen, *Staphylococcus aureus* (S.a.), is controversial. This pathogen is resistant to antibacterial mechanisms of phagocytes and can use them as a replicative niche [247]. An in vivo role for Nox2 during infection with S.a. was suggested in a pulmonary infection model [248]. While Nox2-deficent mice showed increased bacterial burden, Nox2-deficient BMDM in vitro did not differ in their bacterial load. This is not surprising because BMDM express much less Nox2 and show strongly reduced extracellular ROS production as compared to tissue macrophages such as PM [58]. Additionally, no ROS measurements were performed, therefore if and how the antibacterial effect of Nox2-derived ROS observed in vivo is due to ROS production by macrophages remains to be elucidated. In line with that, another study also showed that Nox2-deficient BMDM did not differ in bacterial load from wild type (WT) BMDM after S.a. infection [249]. However, treatment of WT or Nox2-deficient BMDM with the global ROS scavenger N-acetyl-cysteine (NAC) or the flavoprotein inhibitor DPI (see Section 6 and Section 7) resulted in reduced total cellular ROS levels, reduced bacterial load and reduced induction of the Inositol-Requiring Enzyme 1α (IRE1), which is a sensor of endoplasmatic reticulum stress. Since IRE1-deficient BMDM showed reduced bacterial burden and total cellular ROS levels, a ROS-mediated antibacterial role for IRE1 was suggested. The sources of this ROS burst were not analyzed, but since DPI inhibits not only Nox enzymes but also influences the ETC [63,250,251] (see Section 7) and therefore mtROS production, both other Nox isoforms as well as mitochondria could be involved. In a continuative study by the same group, a complex antibacterial role for mitochondria was identified during infection with S.a. [252]. While other studies [61,246,253] have demonstrated that mitochondria can be recruited to bacteria-containing phagosomes and increase antibacterial matrix mtROS production, the study by Abuaita et al. suggested a much more complicated involvement of mitochondria during S.a. infection. S.a.-infected RAW cells showed increased matrix mtROS production and treatment with Necrox-5, a rarely used mitochondrial matrix ROS scavenger, did not only reduce matrix mtROS levels but also increased bacterial burden indicating a role for matrix mtROS for defense against S.a. TLR-mediated signaling can induce mtROS production [30,61], but in contrast TLR2/4/9-deficient RAW cells infected with S.a. showed unaltered matrix mtROS production, while bacterial burden, however, was reduced. Abuaita et al. suggested that TLR signaling, instead of triggering mtROS production, induces the delivery of mitochondria-derived vesicles containing the mitochondrial matrix protein SOD2 to S.a.-containing phagosomes. In the phagosomal lumen, the delivered SOD2 then enhances conversion of O_2_^●−^ to H_2_O_2_. Unfortunately, ROS levels in S.a.-containing phagosomes were not determined, e.g., with bacteria coupled to a ROS probe [246] or with cell-impermeable extracellular ROS probes (see Section 5.2) [30,58,99,254], and therefore clear evidence is missing that phagosomal ROS production really was increased by the SOD2 delivered into the phagosomes through mitochondria-derived vesicles. Experiments with SOD2-deficient macrophages could also have strengthened this suggestion. It remains to be investigated if this very complex pathway to enhance antibacterial ROS production is only induced by S.a. or also by other pathogens and whether it contributes to antibacterial immunity in vivo. At least for L.m., extracellular ROS production is completely independent of TLR signaling [58] excluding a role for TLR-mediated delivery of SOD2 into L.m.-containing phagosomes. A clear role for either Nox2-derived ROS or mtROS in direct antibacterial defense of macrophages against S.a. therefore remains not fully characterized. Future studies with well-established ROS measurements and solid genetic evidence will be needed to clarify this topic.

Like S.a., *Mycobacterium tuberculosis* (Mtb) uses phagosomes of macrophages as proliferative niche. Mechanisms to survive in phagosomes are the inhibition of fusion with lysosomes and avoiding to be targeted by xenophagy [255]. Through its virulence factor CpsA, Mtb also actively inhibits Nox2 recruitment to phagosomes. By preventing Nox2-derived ROS production, Mtb effectively avoids being targeted and killed by LAP [241,256]. Since Nox2-derived ROS production is effectively inhibited and Mtb escapes into the cytosol, macrophages have to activate other cytosolic ROS sources to inactivate this pathogen. An antibacterial role for cytosolic mtROS against Mtb was investigated in two in vivo studies in zebrafish [129,257]. Mtb infection alone was not sufficient to increase total cellular ROS levels or to induce matrix mtROS production in zebrafish macrophages. Priming with TNF, however, induced production of antibacterial matrix mtROS by two converging signaling pathways. Firstly, TNF mediated induction of matrix mtROS production by ryanoid-receptor-receptor-interacting serine/threonine-protein kinase 1 (RIPK1)-RIPK3-mediated calcium overload of mitochondria [257]. Secondly, since matrix mtROS cannot escape the matrix of intact mitochondria [127,128,129], TNF opened the mPTP through a signaling pathway involving RIPK1, RIPK3 and mixed lineage kinase domain like pseudokinase (MLKL). The early effect of cytosolic mtROS released by the opened mPTP led to killing of Mtb, but later on it resulted in necroptosis of macrophages [129]. Ambiguously, the authors suggested mtROS production into phagosomes, however, this was not shown in the two studies. As mentioned above, Mtb evades degradation in the phagosome and quickly escapes into the cytosol, where, as nicely shown by the two articles, the cytosolic mtROS contributed to their killing. In contrast to the studies in zebrafish, Kim and colleagues did not find any antibacterial role for matrix mtROS during Mtb infection in BMDM [258]. They also observed only a minor increase in matrix mtROS production by infection alone. However, this was strongly increased in BMDM deficient for the mitochondrial deacetylase sirtuin 3 (SIRT3). Deficiency for SIRT3 resulted in mitochondria with swollen and disrupted cristae, and damaged mitochondria are known to produce high amounts of matrix mtROS [127,129,130,131]. However, these damage-induced matrix mtROS in SIRT3-deficicent BMDM were only a side effect and did not contribute to direct antibacterial defense. Instead, SIRT3 induced antibacterial xenophagy of Mtb that had escaped from the phagosome. In summary, these studies show that since Mtb efficiently escapes phagosomal degradation and inactivation via Nox2-derived ROS production, macrophages activate excessive mtROS production while going into cell death to minimize the spreading of Mtb infection (see Figure 5).

*Salmonella typhimurium* (S.t.), a pathogen responsible for severe food-borne illness and a major cause of diarrheal diseases [259], also can evade degradation in the phagosome and replicate in macrophages [260]. The importance of ROS during infection with S.t. is highlighted by the strongly increased susceptibility of Nox2-deficient mice to S.t. infection [261]. The antibacterial role of Nox2-derived ROS produced by macrophages during S.t. infection was investigated by many studies [262,263,264], however, S.t. has a big repertoire to counter and even use oxidative stress for their benefit [265]. While the destructive potential of neutrophils by OCl^−^ generation via myeloperoxidase (MPO) is clear-cut [266], macrophages seem to struggle in a balanced fight with S.t., resulting in some S.t. being killed, while the other successfully manage to remodel phagolysosomes into a replicative niche [266,267].

Beyond Nox2, mitochondria were introduced as new players for antibacterial ROS production by West and colleagues [61]. They demonstrated for the first time that S.t. infection of BMDM induced mitochondrial recruitment and matrix mtROS production near phagosomes in a TLR-myeloid differentiation primary response 88 (MyD88)-TNF receptor associated factor 6 (TRAF6)-dependent manner. The matrix ROS production then contributed to the inactivation of S.t. Another study also observed clustering of mitochondria around phagosomes and induction of matrix mtROS production in S.t.-infected zebrafish macrophages [253]. Clustering of mitochondria around phagosomes and mtROS production both were dependent on the immune-responsive gene 1 (IRG1) protein, a 2-methylcitrate dehydratase. It was suggested that IRG1 fuels the citrate cycle and thereby enhances electron flow through the ETC and in consequence also increases matrix mtROS production. Accordingly, in IRG1-deficient macrophages mitochondrial clustering and matrix mtROS production were not induced and bacterial burden was increased. Like Mtb, S.t. has a plethora of methods to survive Nox2-dependent ROS production in the phagosome. However, while Mtb escapes into the cytosol, S.t. remains in the phagosome. In response, macrophages recruit mitochondria to the phagosome to increase the ROS production in the phagosome finally overcoming the antioxidative capacities of S.t. (see Figure 5).

Garaude et al. also investigated the rearrangement of ETC complexes during infection with Gram-negative bacteria [268]. In BMDM infected with E.c., mitochondria showed reduced complex I activity but enhanced complex II activity. While induction of matrix mtROS with rotenone [269], a commonly used positive control for matrix mtROS measurements (see Section 5.2 and Section 7) [30,270,271], was still possible, E.c. infection induced no matrix mtROS production suggesting that the activity shift to complex II did not result in enhanced matrix mtROS production. By contrast, extracellular ROS production induced by E.c infection was by Nox2 since Nox2-deficient BMDM completely failed to produce extracellular ROS after infection. Notably, Nox2-deficient BMDM also showed reduced complex II activity suggesting a signaling function of Nox2-derived extracellular ROS in ETC rearrangement. The mitochondria-located feline gardner-rasheed tyrosine kinase (FGR) was critical for increasing complex II activity, since FGR-deficient BMDM showed reduced complex II activity. Unfortunately, mechanistic insights if and how Nox2-derived extracellular ROS reach and regulate FGR inside mitochondria are missing leaving these two main findings in a correlative relation. Additionally, whether complex II has direct antibacterial functions was not addressed.

A recent study also investigated the role of mitochondrial FGR in macrophages in the context of obesity [272]. In contrast to the study by Garaude et al., matrix mtROS and not Nox2-derived ROS were crucial for activation of FGR, supporting the observation of other studies that ROS have to be produced in direct vicinity of the regulated target and do not excessively saturate the cell until they hit their target at random [29,30,37,61,125,273].

#### 4.1.2. ROS vs. Parasites

While Nox2 and mitochondria so far are the only ROS sources identified to be activated by macrophages upon bacterial challenge, several other Nox enzymes were suggested to be involved in defense against infection with protozoan parasites.

*Toxoplasma gondii* (T.g.) is such a protozoan parasite and responsible for the disease toxoplasmosis, affecting one third of the human population [274,275]. Macrophages play a central role in immune defense against this pathogen [276]. While antimicrobial ROS production in general was one of the first responses identified in T.g.-infected macrophages [277,278], the relative contributions of different ROS sources are still unclear. An in vivo role for ROS production by Nox1 and Nox2 was suggested because Nox1- or Nox2-deficient mice infected with T.g. showed an increased parasitic burden [279]. T.g. infection also increased total cellular ROS levels in WT BMDM in vitro, which was abolished in Nox1- or Nox2-deficient BMDM, while parasitic burden was increased, suggesting a role for Nox1 or Nox2 as important sources of ROS in antiparasitic defense. However, ROS levels were only minimally reduced in Nox1- or Nox2-deficient BMDM in comparison to WT BMDM, which hardly explain the strongly reduced parasitic burden observed in vitro and in vivo. Since mtROS, which can contribute to extracellular and cytosolic ROS production, were not investigated, a role for their antiparasitic defense cannot be ruled out. Of note, another study showed that Nox2-deficient BMDM infected with T.g. showed no alterations in parasitic burden [280]. Nox4 was suggested as another source of antiparasitic ROS, since Nox4-deficient mice showed fewer parasitic cysts and parasitic burden was increased in Nox4-deficicent BMDM. Nox4 is mainly found on intracellular organelles, such as the ER and mitochondria [37,83,281,282], while evidence for Nox4 localization at phagosomal membranes is missing so far. Since no ROS measurements in Nox2- or Nox4-deficient cells were performed, the location (T.g.-containing phagosome or other cellular organelles) and the function (directly or indirectly antiparasitic) of Nox4-derived ROS remained elusive. The ability of *Toxoplasma* parasites to detoxify phagosomal ROS was demonstrated with the evolutionary related *Toxoplasma cruzii* in phagosomes of J774A.1 macrophage-like cells (J774) and BMDM [283]. No ROS measurements with classical probes were performed, but elegant and technically challenging direct measurements of O_2_^●−^ in the parasitic cytosol and the phagosomes were performed in WT and Nox2-deficient J774 cells. The superoxide dismutase Fe-SODB in the parasitic cytosol was identified as the main defense mechanism against antiparasitic ROS. Fe-SODB is conserved among *Toxoplasma* parasites suggesting that also other *Toxoplasma* strains may possess this ability, questioning the still controversially discussed direct antiparasitic role for ROS in T.g.-infected macrophages.

Protozoan parasites of the genus *Leishmania* cause the tropical disease Leishamaniasis [284]. Neutrophils phagocytose and kill 80–90% of the invading *Leishmania* parasites [285,286]. Macrophages also participate in removal of parasites, however, their role is of dual nature, similar to S.t. infection. Macrophages manage to kill some of the phagocytozed parasites but also constitute a replicative niche [287,288]. At least one *Leishmania* subspecies, *Leishmania major*, actively impairs recruitment of Nox2 to phagosomes, which leads to reduced extracellular ROS production and reduced activation of LAP [241,289]. It is tempting to speculate that all *Leishmania* subspecies are capable of reducing direct antiparasitic ROS production through inhibition of Nox2 recruitment. *Leishmania amazonensis* infection in J774 cells induced an increase in Nox2 protein levels and total cellular ROS levels [290]. However, no Nox2-deficient macrophages were used to corroborate this suggestion. ROS production was inhibited with DPI, which is not specific for Nox2 (see Section 7), leaving the question of the precise ROS source unanswered. While the antimicrobial role for ROS against parasites in general is unquestioned, the involved ROS sources remain controversial and need further investigations.

#### 4.1.3. ROS vs. Viruses

While many studies focused on bacterial and parasitic infections, studies investigating the direct role of ROS during viral infection of macrophages are rare.

A conclusive study demonstrated a role for Nox2 during viral infection of macrophages [291]. The authors showed that macrophages infected with single-stranded RNA viruses such as influenza A viruses, respiratory syncytial virus, rhinovirus, Dengue virus or HIV or the DNA viruses vaccinia virus and herpes simplex virus showed increased endosomal ROS production, which was abolished in Nox2-deficicent cells. ROS production by Nox2 was induced by virus recognition through TLR7 and subsequent protein kinase C (PKC) activation. Mechanistically, TLR7-induced Nox2-derived endosomal ROS negatively regulated TLR7 on the cysteine residue Cys^98^ in a negative feedback loop and therefore dampened the antiviral response. This study showed that Nox2-derived ROS had detrimental consequences for the host by negatively influencing the antiviral response. Accordingly, in vivo Nox2 inhibition protected mice from influenza A virus infection. Moreover, several other studies also described a rather exacerbating role for Nox-derived ROS during virus infection in various cell types and disease settings [292,293,294,295].

A role for mtROS during virus infection in vivo was investigated in a study from the same group [296]. Administration of the matrix mtROS scavenger mitoTEMPO to mice during infection with influenza A virus decreased inflammation of the airways, body weight loss and mortality. Not surprisingly, macrophage populations isolated from treated mice by bronchoalvelolar lavage showed reduced levels of matrix mtROS. A mechanistic explanation how matrix mtROS modulate the antiviral response, as nicely investigated for Nox2-dervied ROS in the previous study [291] was unfortunately not provided.

Surprisingly, studies investigating a direct antiviral role for ROS in macrophages, e.g., by induction of oxidative damage to the viral genome [297] or by modulation of autophagic pathways needed for genome replication, packaging or release of viral particles [298,299,300,301] are largely missing. Hopefully, also in regard of the corona pandemic [302,303,304], future studies will investigate this interesting research field.

### 4.2. Immune-Regulatory Functions of ROS

The role of ROS as signaling molecules in general [29,305] and in macrophages in particular [42] has been demonstrated by many studies. If and how ROS regulate the pro- and/or anti-inflammatory response of macrophages depends on various factors, such as the source and compartmentalization of the ROS and the respective stimulus [36,234,306]. However, studies that have investigated ROS-dependent cellular signaling in infected macrophages are surprisingly scarce.

#### 4.2.1. Immune-Regulatory Functions of mtROS

While the direct antimicrobial role of mtROS has been intensively investigated, their role in the proinflammatory response of infected macrophages remained an open question and the underlying molecular mechanisms through which mtROS mechanistically regulate proinflammatory signaling remained elusive. In a recent study, we have elucidated for the first time the role of mtROS in proinflammatory signaling of macrophages during bacterial infection [30]. Infection of PM with L.m. induced the production of extracellular ROS (see Section 5.2), which was strictly Nox2-dependent, and the production of cytosolic ROS (see Section 5.2), which was completely independent not only of Nox2 but also of all other Nox isoforms. PM deficient for Nox1, Nox4, Duox1, and Duox2 showed neither reduced extracellular nor cytosolic ROS production. We could also show that bacterial infection induced cytosolic mtROS production via complex III of the ETC in a TLR-MyD88-TRAF6-dependent way. These cytosolic mtROS were crucial for proinflammatory signaling. Notably, Nox2-derived extracellular ROS production was completely independent of TLR signaling and instead depended on the integrin Mac-1, which was of crucial importance for induction of the highly antimicrobial pathway of LAP [58,241,243]. In contrast to infection with other pathogens, like Mtb or S.t. [129,131], L.m. infection did not induce matrix mtROS production in PM. To further exclude a role for matrix mtROS in proinflammatory signaling, matrix O_2_^●−^ was scavenged with mitoTEMPO and H_2_O_2_ was removed by overexpression of catalase in the mitochondrial matrix but in both cases neither cytosolic ROS levels nor cytokine secretion were altered. Infections with pathogens, which cause mitochondrial damage also cause matrix mtROS production [129,131] but in the case of L.m. contrasting observations were reported. While some studies reported L.m.-induced damage in thioglycolate-elicited PM and in the cancer-derived epithelial cell line HeLa. [307,308], L.m.-induced damage to mitochondria was not observed in BMDM and naïve PM [30,309] indicating that differences in cell type, macrophage isolation, L.m. strains, multiplicity of infection, and duration of infection seem to highly influence whether L.m. induce mitochondrial damage and subsequent matrix mtROS production or not. Moreover, from intact mitochondria matrix mtROS cannot reach the cytosol or other compartments [38,122,127,129,131,310].

Of note, our study provided a mechanistic target for mtROS-mediated proinflammatory signaling, namely the IκB kinase γ/NF-kappa-B essential modulator (IKKγ/NEMO) subunit of the IKK complex, which is essential for initiation of the proinflammatory pathways leading to cytokine secretion (see Figure 6) [311,312]. A previous study showed that NEMO dimerization via disulfide bonds and subsequent NF-κB signaling after TNF stimulation in mouse embryonic fibroblasts (MEF) depended on two redox-sensitive cysteines, Cys^54^ and Cys^374^, [40] and, indeed, expression of a redox-insensitive NEMO mutant in PM, BMDM and MEF through transfection of in vitro-generated mRNA [123] completely abolished disulfide linkage of NEMO, proinflammatory signaling and cytokine secretion in response to bacterial infection. So far, our work provides the only mechanistic explanation how ROS in general and mtROS in particular modulate proinflammatory signaling of infected macrophages and it is tempting to speculate that NEMO also represents the redox-target for ROS produced in other inflammatory scenarios (see Figure 6).

Other studies have investigated mtROS-mediated signaling after stimulation with the Gram-negative PAMP LPS instead of bacterial infection. The influence of metformin on mtROS production and mtROS-dependent cytokine secretion was suggested in a study from Kelly and colleagues [313]. Metformin treatment inhibited complex I activity in unstimulated BMDM. After stimulation with LPS, metformin-treated BMDM showed reduced total cellular ROS levels (see Section 5.1), decreased IL-1β cleavage, decreased TNF secretion yet enhanced secretion of the anti-inflammatory cytokine IL-10, suggesting a regulating role for ROS during proinflammatory signaling, which can be inhibited via Metformin treatment. However, if and how metformin treatment influences matrix mtROS production was not investigated as no matrix mtROS measurements were performed. Moreover, treatment with MitoQ, a substance known to scavenge matrix mtROS (see Section 6) [314,315], gave contrasting results: only IL-1β cleavage was reduced, whereas TNF and IL-10 levels remained unaltered. Therefore, a direct proof that the total cellular ROS levels that were reduced after metformin treatment originated from mitochondria is missing. In consequence, no connection between metformin treatment and mtROS production was shown, leaving reduced complex I activity, reduced total cellular ROS levels and altered cytokine levels after metformin treatment as purely correlative statements. Another study investigated the interplay between matrix mtROS and TNF receptor 1 (TNFR1)-mediated signaling [63]. TNFR1-deficient macrophages stimulated with LPS showed enhanced secretion of IL-6, TNF and IL-1β. Cytokine secretion was not altered in Nox2-deficient or Nox2/TNFR1 double-deficient macrophages. Moreover, macrophages lacking the p22^phox^ subunit showed cytokine secretion similar to Nox2-deficient macrophages, nicely showing that Nox enzymes were dispensable for cytokine secretion after LPS stimulation. Not surprisingly, the NLRP3 inflammasome was, with exception for IL-1β, not involved in cytokine secretion as NLRP3-, Caspase-1- and Caspase 1l-deficient macrophages showed normal secretion of TNF and IL-6 (see Section 4.3). Unfortunately, experiments concerning involvement of matrix mtROS were only performed in MEF and not in macrophages. MEF treated with LPS and MitoQ showed reduced levels of IL-6, which, unfortunately, is the only connection between mtROS and cytokine secretion investigated in this study. Moreover, no mechanistic explanation was provided for how mtROS regulate activation of the NLRP3-inflammasome (see Section 4.3).

#### 4.2.2. Regulatory Functions of Nox-Derived ROS

Involvement of ROS generated by Nox2 in proinflammatory signaling was suggested by several studies [316,317]. While Nox enzymes were not involved in proinflammatory signaling in response to bacterial infection or LPS of macrophages [30,63], Allan and colleagues demonstrated an important role for Nox2-derived extracellular ROS in antigen presentation [232]. Phagocytosis of immunoglobulin G (IgG)-opsonized particles induced extracellular ROS production, which was abolished in Nox2-deficient BMDM. Nox2-dervied ROS inactivated in a redox-dependent manner the phagosomal cathepsins L and S, thereby reducing uncoordinated proteolysis of engulfed peptides and increasing proper presentation as antigens by major histocompatibility complex (MHC)-class II molecules (see Figure 6). Accordingly, Nox2 deficiency of antigen-presenting BMDM resulted in increased cleavage of several investigated phagocytosed antigens and reduced MHC-class II-dependent presentation. In vivo, Nox2-deficient animals showed partial protection from an antigen-induced model of autoimmune encephalomyelitis, corroborating the in vitro findings. A hyperinflammatory role for Nox2-derived ROS was also investigated in vivo [318]. Sterile inflammation induced via intraperitoneal zymosan injection led to enhanced mortality and general symptoms of systemic inflammation in Nox2-deficient animals. Polymorphnuclear leukocyte (PMN) recruitment to the peritoneal cavity was similar in WT and Nox2-deficient animals, however, the latter showed prolonged persistence of PMN and the recruited cells showed enhanced secretion of chemokines. Bronchoalveolar lavage analysis revealed also increased recruitment of PMN to the lung in Nox2-deficient animals as well as more thrombi and hemorrhage. Serum levels of several cytokines and chemokines were increased and prolonged in Nox2-deficient animals. However, a mechanistic explanation for the hyperinflammatory phenotype caused by the lack of Nox2-derived ROS was not given.

A conceiving study demonstrated a role for Nox1-derived ROS in a hepatocellular carcinoma model in mice and patients lacking Nox1 [319]. Nox1-deficient, but not Nox4-deficient, animals showed strongly reduced tumor numbers and size and reduced liver damage. Nox1/Nox4 double-deficient mice resembled Nox1-deficient mice excluding an interconnection between the two Nox enzymes. Total ROS levels in tumor tissue, tumor cell proliferation and hepatic levels of the proinflammatory cytokines IL-6 and TNF were reduced after chemically induced liver damage in Nox1-deficient animals. Liver macrophages were identified via cell-type specific Nox1 deletion as promoters of a proinflammatory liver environment, hepatocyte proliferation, tumor development and liver damage, while Nox1 deletion in hepatocytes and bile duct cells had no effect. Mechanistically, recognition of damage-associated molecular patterns (DAMP) released from damaged hepatocytes in vitro increased total cellular ROS levels (see also Section 5.1) and induced extracellular signal-regulated kinases 1/2 (ERK1/2)-mediated expression of IL-6 and TNF in BMDM. Nox1-deficient BMDM showed no ROS production and no expression of cytokines, nicely corroborating the in vivo findings. While the role for proinflammatory signaling of Nox1-derived ROS was clearly shown, no mechanistic redox-sensitive target was identified for Nox1-derived ROS on the cellular level. It is tempting to speculate, since cytosolic mtROS mediate ERK1/2-depened proinflammatory signaling in infected macrophages via NEMO [30], that also in this setting Nox1-derived ROS may act via this mechanism. Recently, Liu and colleagues elegantly demonstrated through in vivo ROS measurements a role for Nox1 during LPS-induced intestinal inflammation [320]. In contrast to WT mice, Nox1-deficient mice showed no detectable ROS production in the ileum after LPS treatment. Nox1-derived ROS production was necessary for upregulation of inducible NO synthase (iNOS), but not vice versa. In both Nox1-and iNOS-deficient animals, the intestinal barrier was impaired whereas cell death and inflammation in the ileum were unaffected. Nox1-deficient ileal macrophages, but not epithelial cells, showed reduced iNOS expression after LPS stimulation. Moreover, secretion of matrix metalloproteinase 9 (MMP9), a protein catalyzing several functions such as cleavage of tight junctions [321,322], was reduced in Nox1-deficient macrophages in a p38 mitogen-activated protein kinase (p38)-dependent manner. Accordingly, Nox1-deficient animals showed reduced cleavage of tight junctions in the intestine suggesting a critical role for Nox1-derived ROS in regulation of iNOS expression and MMP9 secretion by intestinal macrophages and subsequent cleavage of ileal tight junctions. However, again no direct target for Nox1-derived ROS was identified.

### 4.3. ROS and Inflammasomes

Inflammasomes are cytosolic high-molecular weight complexes that are activated by various different stimuli such as infection, PAMPs and DAMPs. In most cases, inflammasome activation results in assembly of the adaptor protein apoptosis-associated speck-like protein containing a CARD (ASC), recruitment and autocatalytic activation of caspase-1 and finally release of IL-1β and IL-18. Initiation of pyroptosis, a regulated lytic cell death pathway, can be a consequence of inflammasome activation but is not obligatory. Five receptor proteins leading to inflammasome assembly were identified. The canonical inflammasomes are activated by the nucleotide-binding oligomerization domain (NOD) of the leucine-rich repeat (LRR)-containing protein (NLR) family members NLRP1, NLRP3, NLRC4, NLRP6, NLRP7, NLRP9b, the absent in melanoma 2 (AIM2)-like receptor (ALR) AIM2 and Pyrin [323,324,325,326]. The noncanonical inflammasome, which activates caspase-11 in mice and caspase-4 and/or caspase-5 in humans, is activated through direct recognition of LPS in the cytosol [323]. Other inflammasomes, such as NLRP2, NLRC5, NLRP12, retinoic acid-inducible gene I (RIG-I), and IFNγ-inducible protein 16 (IFI16) also activate caspase-1 but the underlying mechanisms are not yet fully understood [324,327,328,329,330,331]. The crucial role for inflammasome activation during infection of macrophages is well established and has been excellently reviewed elsewhere [324,332,333]. It is also clear that ROS are necessary for inflammasome activation. The underlying molecular mechanisms of ROS-mediated inflammasome activation, however, remain poorly defined. Here, we will focus on studies that investigated the role of ROS during inflammasome activation in macrophages.

A role for ROS in general for NLRP3 inflammasome activation was demonstrated in an elegant study by Zhou and colleagues [334] showing that the NLRP3 inflammasome can be activated through direct treatment of the human leukemia-derived macrophage-like cell line THP-1 (THP-1) with H_2_O_2_. High concentrations of H_2_O_2_ (10 mM) disrupted the interaction of the thioredoxin-interacting protein (TXNIP) with thioredoxin leading to its interaction with NLRP3 resulting in caspase-1 activation and finally IL-1β cleavage. Phagocytosis of uric acid crystals, as inflammasome-activating stimulus, also led to increase in total cellular ROS levels (see also Section 5.1) and dissociation of TXNIP from thioredoxin. In TXNIP-deficient BMDM, cleavage of caspase-1 and IL-1β was strongly reduced. The potential sources of ROS necessary for NLRP3 inflammasome activation, however, are controversially discussed.

#### 4.3.1. Nox-Derived ROS in Inflammasome Activation

Three studies showed independently that Nox2 is completely dispensable for NLRP3 inflammasome activation in human macrophages. In all studies, peripheral blood-derived monocytes (PBMC) from patients with CGD were analyzed. The studies covered patients lacking Nox2 subunits, e.g., gp91^phox^ and p47^phox^, or the common catalytic subunit for Nox1-Nox4, p22^phox^. Whole blood samples [335] or isolated PBMC [335,336,337] showed no alterations in levels of active caspase-1 or IL-1β/IL18 secretion after treatment with typical inflammasome-activating stimuli, such as LPS/ATP, silica or uric acid crystals. Of note, not reduced but elevated IL-1β/IL-18 levels were observed in PBMC from patients lacking gp91^phox^ [337]. Previous findings suggested a role for Nox2-derived ROS during NLRP3 inflammasome activation triggered by silica and asbestos particles [338] and by hemozoin, a crystalline virulence factor secreted by malaria-inducing *Plasmodium* species [339]. While the in vivo relevance NLRP3 inflammasome activation for the innate immune response after the stimuli is clear-cut and well investigated, the findings, which suggested Nox2 as ROS source were based on experiments with THP-1 cells treated with DPI. The use of DPI as specific Nox inhibitor is highly questioned [340,341] and usage of Nox-deficient cells for proof of Nox-derived ROS is recommended [342] (see Section 7).

In contrast to humans, a role for Nox4 in activation of the NLRP3 inflammasome was suggested in mice during infection [343]. Nox4-deficient animals infected with *Staphylococcus pneumonia* showed reduced IL-1β levels in whole lung lysates. WT BMDM treated with LPS/ATP showed enhanced expression of Nox4, whereas Nox4-deficient BMDM treated with ATP/LPS showed reduced levels of IL-1β and reduced ASC speck formation. Nox4-deficient BMDM also showed reduced fatty acid uptake into mitochondria, reduced fatty acid oxidation and reduced levels of carnitine palmitoyltransferase 1 (CPT1), a key enzyme for fatty acid oxidation. However, no mechanism for the role of Nox4-derived ROS was shown. Location of Nox4 to mitochondria was demonstrated by Western blot analysis of cellular fractions and Nox4-deficient BMDM showed reduced matrix mtROS production after stimulation with LPS and nigericin. However, if and how Nox4-dependent matrix mtROS production regulates inflammasome activation in a CPT1-dependend manner was not investigated. Therefore, in mice, it also remains elusive if Nox-derived ROS directly activate the inflammasome in infected macrophages.

#### 4.3.2. mtROS in Inflammasome Activation

While the role for Nox-derived ROS for NLRP3 inflammasome activation in infected mouse macrophages still needs further investigations, a subsequent study from Zhou et al. identified mitochondria as source of the ROS that are required for NLRP3 inflammasome activation in human and mouse macrophages [273]. Zhou and colleagues showed that induction of matrix mtROS production by complex I of the ETC is necessary for IL-1β secretion by THP-1 cells. Additionally, in Lrp3-deficient BMDM, which fail to induce mitophagy and accumulate damaged mitochondria, production of matrix mtROS and subsequent IL-1β secretion was increased. As one of the few studies which investigated the roles of ROS in macrophages, the authors here show nicely that the target of redox regulation, the thioredoxin-interacting protein (TXNIP), had to be recruited from the cytosol to the source of the ROS, i.e., the mitochondria, where matrix mtROS-dependent oxidation and dissociation from thioredoxin took place (see Figure 6). This study further strengthens the opinion that ROS do not saturate the cell but that either ROS are produced in the vicinity of their target or the redox target is recruited to the site of ROS production [33]. Despite the fact that cellular compartments highly differ in their redox regulation and signaling [36,234,344,345], studies that have highlighted the importance of localized ROS production are disappointingly rare [30,37,61,125,272,273].

The ROS-mediated regulation of the NLRP3 inflammasome in macrophages is extensively investigated, but nearly nothing is known about the source and the role of ROS for activation of other inflammasomes. One study showed that matrix mtROS production was also crucial for activation of the AIM2 inflammasome in BMDM infected with *Francisella tularensis* [346]. Global scavenging of ROS with NAC or matrix mtROS scavenging with mitoTEMPO resulted in reduced Caspase-1 activation and IL-1β secretion, while Nox2-deficiency had no effect.

Like in nearly all fields of ROS-mediated immunity in infected macrophages, the general role of ROS for inflammasome activation is unquestioned. While identification of ROS sources took momentum and mitochondria as main ROS source for inflammasome activation became evident, only two studies so far identified a redox-dependent target necessary for NLRP3 inflammasome activation, namely TXNIP [273,334]. Future studies will have to evaluate if other inflammasomes are also activated in a direct ROS-dependent manner in infected macrophages and carefully investigate the activated sources and the targets of ROS.

## 5. ROS Probes

A plethora of ROS probes are available and with ongoing development the choices are frequently widened. Most precise ROS measurements concerning compatibility and specificity for ROS subspecies can be achieved with genetically modified cells, which express the ROS probe of choice in the cellular compartment of choice. Examples are the HyPer reporter family and reduction-oxidation sensitive green fluorescent protein 2- OSBP Related Protein 1 (roGFP2-Orp1) [347,348,349,350,351]. However, since these approaches require genetical modifications, which are challenging and time consuming to establish and are often restricted to cell lines, we will focus in this review on the commonly used and commercially available ROS probes, which also allow ROS measurements in ex vivo cells and point to an excellent summarization on the topic for further reading [352].

The listed ROS probes either emit chemiluminescence via a ROS-consuming reaction or emit fluorescence after reaction with ROS. All ROS probes listed are usable in a plate reader and therefore, in terms of used media, one must consider that every type of pH indicator (e.g., phenol red) or serum (e.g., normal mouse serum) leads to falsification of both, luminescence and fluorescence signals and must be avoided.

### 5.1. Diffusable ROS Probes (Total Cellular ROS Detection)

Diffusible ROS probes are not retained or targeted to a specific cellular compartment, however, some show specificity for ROS subspecies. Therefore, with the ROS probes listed below, only total cellular ROS levels in cells can be measured (see Figure 7).

Luminol is a luminophore that reacts with all types of ROS. In the presence of an endogenous or exogenous peroxidase, such as horseradish peroxidase (HRP), and ROS, luminol is oxidized to 3-aminophtalate. This molecule is in an energetically excited state and emits light when returning to ground state. The generated chemiluminescence represents a quantitative value for the amount of ROS generated [353,354]. Since this probe is diffusible, both extra- and intracellular ROS, i.e., total cellular ROS levels are detected (see Figure 7) [254,355].

Lucigenin was originally described as probe for O_2_^●−^ detection [356]. However, other studies demonstrated that this probe does not directly interact with O_2_^●−^, but instead with cytochrome c and XO [357], increases NADPH oxidation and binds to NO with low affinity [358]. Since a big repertoire of other ROS probes is available, lucigenin should therefore not be used for ROS detection.

2′,7′-Dichlordihydrofluorescein-diacetat (H_2_DCF-DA) is a cell-permeable derivative of fluorescein and one of the most commonly used ROS probes. After its diffusion into the cytosol, the two acetate groups are cleaved by cellular esterases generating 2′,7′-dichlordihydrofluorescein (H_2_DCF). This molecule can now be oxidized by ROS to generate the highly green fluorescent 2′,7′-dichlorofluorescein (DCF). H_2_DCF-DA is often referred to as intracellular ROS probe. However, even after cleavage of the acetate groups, it remains diffusible and therefore does not only remain in the cytosol but also reaches cell organelles and diffuses back into the extracellular space. Therefore, like with luminol, only total cellular ROS levels can be detected with this probe (see Figure 7) [355,359].

Dihydroethidium (DHE) is a red fluorescent probe that rather specifically reacts with cellular O_2_^●^^−^, resulting in the two fluorescent products ethidium and 2-hydroxyethidium. Only 2-hydroxyethidium is generated after reaction with O_2_^●−^, while ethidium is formed by nonspecific redox reactions. However, DHE easily crosses cellular membranes and therefore can be oxidized by O_2_^●−^ anywhere in and outside of the cell [360]. Therefore with this probe only total cellular O_2_^●−^ production can be detected. Notably, after oxidation both products will bind to DNA, which increases their fluorescence. This might lead to false interpretations regarding the localization of ROS production, namely all DNA containing organelles, like the nucleus and mitochondria, when immunofluorescence microscopy is used to analyze ROS production (see Figure 7).

CellROX oxidative stress reagents are different cell-permeable fluorescent probes that are mainly used as indicators for oxidative stress. The different CellROX probes are diffusible and cannot differ between cellular compartments and ROS subspecies, but vary greatly in usage conditions, like live cell compatibility, resistance to detergents, fixation capabilities, and emitted fluorescence spectra. However, since it can be coupled to bacteria, compartment-specific ROS measurement of phagocytosed bacteria in the phagosome can be performed [246].

### 5.2. Nondiffusable ROS Probes (Compartment-Specific ROS Detection)

In contrast to the previous listed ROS probes, the following list contains either cell-impermeable ROS probes or probes targeted or retained to a specific cellular compartment. Some but not all probes additionally detect specific ROS subspecies. Therefore, these ROS probes offer a more detailed and specific approach to analyze ROS production in cells (see Figure 7).

Isoluminol is a derivative of the chemiluminescent luminol with the amino group placed on a more exposed position on the molecule. This modification makes isoluminol cell-impermeable. This makes it ideal for exclusive measurement of extracellular ROS, which also includes ROS produced into the lumen of endosomes and phagosomes [30,58,99,254,353,361]. However, the probe cannot differ between different ROS subspecies. A recommended positive control for induction of extracellular ROS in macrophages is PMA [30,58,99].

Amplex Red also is cell-impermeable but, in contrast to isoluminol, detects specifically H_2_O_2_ [362]. In the presence of HRP and H_2_O_2_, Amplex Red is oxidized to red fluorescent 10-acetyl-3,7-dihydroxypenoxazine [363]. In consequence, this probe is optimal for determination of specifically extracellular generation of H_2_O_2_.

5-(and -6)-Carboxy-2′,7′-dihydrochlorofluorescein-diacetat (5/6-Carboxy-DCF) is a derivative of H_2_DCF-DA that contains two additional carboxyl groups that enhance its hydrophilicity and therefore strongly increase its retention in the cytosol. Therefore, in contrast to H_2_DCF-DA, which detects total cellular ROS levels, the green fluorescent 5/6-Carboxy-DCF can be used to specifically detect cytosolic ROS levels (see Figure 7) [30,99,359,364]. However, like isoluminol/luminol, this ROS probe cannot differ between ROS subspecies. As for extracellular ROS detection, PMA can also be used as positive control to induce cytosolic ROS in macrophages.

MitoSOX Red (MitoSOX) is a modified DHE analog attached to a triphenylphosphonium (TPP) group, which leads to accumulation of the probe in the mitochondrial matrix [365,366]. Like DHE itself, MitoSOX specifically detects O_2_^●−^ and emits red fluorescence after the reaction (see Figure 7) [30,99,360]. In contrast to DHE, MitoSOX does so mainly in the mitochondrial matrix, though. Notably, MitoSOX can be readily oxidized by other ROS while diffusing into mitochondria. Presence of Nox2, as an example, leads to a false positive fluorescent MitoSOX Red signals after various stimuli in WT PM, which were abolished in Nox2-deficient PM [30]. A commonly used positive control for induction of matrix mtROS, and therefore MitoSOX, is rotenone (see Section 7) [30,270,271].

## 6. ROS Scavengers

The term “ROS scavenger” broadly describes any chemical or biological molecule, which is capable of detoxifying one or more ROS subspecies by different mechanisms defined by the chemistry of the ROS scavenger and the ROS subspecies [367]. Therefore a ROS scavenger not only scavenges radicals (like ●OH and O_2_^●−^), but can also scavenge nonradicals (like H_2_O_2_) or more than one ROS subspecies in dependency of the chemical structure of the scavenger. Moreover, this should not be mistaken with the term “Antioxidant”, which chemically defines molecules or atoms that can in general reduce an oxidizing substance and are not limited to ROS [66,368]. By this definition, antioxidants can include ROS scavengers, but also other chemicals that reduce for example RNS. Therefore, an antioxidant is not automatically a ROS scavenger and these two terms should not be mixed up. Like ROS probes, a lot of commercially available ROS scavengers are available but differ in their compartment and ROS subspecies specificity making the experimental choice puzzling.

N-acetyl cysteine (NAC) is a commonly used global ROS scavenger, which works extra- and intracellularly. It was long assumed that NAC scavenges oxidants directly through its thiol group and by its ability to act as a source of cysteine for increased GSH synthesis. However, some studies observed that NAC did not restore decreased GSH levels [369] and in vivo-labeling in mice showed that NAC is not used as direct precursor for GSH synthesis in the cell [370]. A recent study suggested that NAC functions as ROS scavenger by triggering intracellular H_2_S and sulfane/sulfur production [371]. As a globally working ROS scavenger, usage of NAC can give only insights into general involvement of ROS in the process of interest. For analysis of compartment-specific functions of ROS, the use of other substances is required.

Tempol (4-Hydroxy-Tempo) is a SOD mimetic and catalyzes the dismutation of O_2_^●−^ into H_2_O_2_. The term ROS scavenger therefore is not accurate, since one ROS subspecies is converted into another. However since O_2_^●−^ is removed, its cellular role can be investigated with this substance [30,372]. Of note, Tempol is cell-permeable and not compartment-specific and hence reduces O_2_^●−^ levels everywhere in the cell [373,374].

Tiron (1,2-dihydroxybenzene-3,5-disulfonate) is a vitamin E analog, cell-permeable and acts as a global O_2_^●−^ scavenger [375,376,377].

Trolox (6-hydroxy-2,5,7,8-tetramethylchroman-2-carboxylic acid), like Tiron, is a water-soluble derivative of vitamin E and globally scavenges ●OOH and ●OOR [378,379].

Ebselen is a heterocyclic seleno-organic compound and exerts glutathione reductase activity. It effectively removes H_2_O_2_ and ONOO^−^ [30,380,381,382].

All of the probes mentioned above are highly diffusible. Therefore, they can be used to determine whether ROS in general or specific ROS subspecies do play a role in a process of interest. They cannot be used to identify the specific compartment in which the ROS exert their function, though. For this, compartment-specific removal of ROS is required, which is possible with the substances described below.

Vitamin C (ascorbic acid and its anion ascorbate) acts as water-soluble reducing agent [383], donating electrons to ROS and therefore acts as ROS eliminating substance [384,385]. Vitamin C and its reduced form dehydroascorbic acid (DHA) are cell-impermeable and imported into the cell via Na^+^ cotransporters in the plasma membrane. While a number of studies have demonstrated extra- and intracellular ROS scavenging activities of vitamin C in different cell lines [386,387], we could exclusively observe extracellular ROS scavenging in macrophages (own unpublished data) and microglia [99]. Cytosolic ROS levels remained unaffected in macrophages and microglia after vitamin C treatment.

Vitamin E is a lipophilic molecule that is exclusively integrated into biological membranes, where it efficiently scavenges radical ROS, mainly ●OOR and therefore acts a major defense against lipid peroxidation in cells [388,389,390]. It does not scavenge nonradical ROS like H_2_O_2_ and does not scavenge ROS in hydrophilic cellular compartments such as the lumen of cell organelles or the cytosol [75].

MitoTEMPO and MitoQ both are targeted to and accumulated in mitochondria due to their TPP group. MitoTEMPO is a combination of the antioxidant TEMPO attached to the TPP group and hence dismutates O_2_^●−^ to H_2_O_2_ in the mitochondrial matrix [314,391], whereas MitoQ is an analog of ubiquinone and prevents lipid peroxidation in mitochondrial membranes [314,315].

In sum, NAC is optimal for a first investigation to analyze, if ROS play a role in the experimental setting at all and we recommend for further experiments Vitamin C for extracellular ROS scavenging, Vitamin E for general and MitoQ for mitochondrial membrane protection against radical ROS and mitoTEMPO for removal of matrix-located O_2_^●−^.

## 7. ROS Source Inhibitors

While ROS scavengers directly remove ROS from the system, but optimally do not influence the sources of ROS production, the use of ROS source inhibitors is an option to block ROS production and analyze possible ROS sources when genetic modifications are not an option. Like in the case of ROS probes and ROS scavengers, there are a lot of ROS source inhibitors commercially available and we will focus on the most commonly used substances observed in many studies so far. For a broader overview, we refer to excellent reviews solely examining this topic [86,392,393,394].

Apocynin was long considered as specific Nox2 inhibitor. However, several studies have shown that this is indeed not the case. Instead, apocynin directly scavenges ROS due to its antioxidant capacities [340,395,396,397,398]. Hence, Nox2 involvement in a process of interest cannot be shown by the sole use of this substance.

DPI (diphenyliodinium), like apocynin, was long considered as specific Nox2 inhibitor. However, due to its chemistry, it unselectively inhibits all flavin-containing enzymes of the cell [393,394,399]. This includes Nox enzymes but also, for example, complex I of the ETC [63,250,251], iNOS [400,401] and XO [394,399] as well as calcium transporters [402]. Therefore, its definition as specific Nox2 inhibitor is outdated and statements about Nox2 involvement based on sole use of this substance are highly questionable.

VAS2870 was the first published inhibitor that resulted from a systematic screening effort for selective NADPH oxidase inhibitors by Tegtmeier and colleagues [394]. VAS2870 is a well validated NADPH oxidase inhibitor [341] as it shows no intrinsic antioxidant activity, does not inhibit other flavoproteins and inhibits NADPH oxidase-mediated ROS production in vitro and in vivo. However, although initial experiments suggested VAS2870 to be Nox2-specific, it is definitely not [341,393,403,404].

GKT136901 and GKT137831 were identified by Laleul and colleagues during a screen for Nox4 inhibitors [405]. Both compounds inhibit Nox1, Nox4 and Nox5 and in higher concentrations also Nox2, Duox1, Duox2 [406,407,408,409], and XO [392,393,408,409]. Notably, GKT136901 also scavenges ONOO^−^, a RNS [410] and therefore works more like an antioxidant than a specific Nox inhibitor. In contrast, so far no direct antioxidative side effects of GKT 137831 were reported [392] highlighting this substance as the most promising inhibitor for Nox enzymes not only in vitro but also in vivo [94,409,411,412,413,414].

Rotenone acts as a strong inhibitor of complex I of the ETC [415,416,417]. The mechanism of action comprises inhibition of electron transfer from the iron-sulfur centers in complex I to ubiquinone [418]. Blockade of rotenone to the binding site of ubiquinone leads to reduced electron flow to complex III and reverse electron flow through complex I into the mitochondrial matrix leading to increased ROS generation in this compartment, namely matrix mtROS [269,416,419,420,421]. Therefore, rotenone is a potent inducer of the generation of matrix mtROS and hence is widely used as positive control for ROS probes targeted to the mitochondrial matrix, such as MitoSOX [30,270,271]. Of note, while increasing matrix mtROS generation, the reduction of electron flow to complex III by rotenone effectively blocks ROS production by this complex and therefore generation of cytosolic mtROS [30,125].

Antimycin A blocks the Qi site of complex III of the ETC [86,422], thereby preventing electrons from entering the complex, which leads to increased leakage of electrons into the IMS. In the IMS, leaked electrons react with O_2_ to generate ROS, which reach the cytosol [30,109,121,125,423]. This may lead to some confusion, as antimycin A is often referred to as inhibitor of mtROS production. Antimycin A acts as inhibitor of the respiratory chain at complex III in terms of ATP production, but its influence on mtROS production is compartment-specific. It induces cytosolic mtROS production instead of inhibiting it, while matrix mtROS production is abolished.

Myxothiazol and stigmatellin, like antimycin A, block electron flow to complex III. However, they do so at the Qo site of the complex, which results in the opposite effect of the one observed with antimycin A. Blockade of complex III with stigmatellin [424] and, with a slightly lower effect, also myxothiazol [425] leads to reduction of cytosolic mtROS production by complex III. Electron leakage into the mitochondrial matrix in consequence is increased, which results in increased generation of matrix mtROS [416].

## 8. Concluding Remarks

The important and versatile functions of ROS in macrophage-mediated immunity are unquestioned and have been demonstrated by many studies. However, only a few studies have performed in-depth ROS analyses, and even fewer have provided mechanistic insights into the redox-regulated targets. Experimental setups that commonly lead to misinterpretation of ROS measurements in macrophages are:usage of only one type of ROS probe without explanation of the rationale behind the choice, such as compartment or ROS subspecies specificity or no specificity at all (“total cellular ROS”);usage of unspecific inhibitors of ROS production or usage of only globally working ROS scavengers;no genetic evidence for the ROS source (especially in case of Nox enzymes);inconsistent use of different stimuli and macrophage cell types for experiments in one study.

In consequence, the highly complex and structured production of ROS in macrophages is often reduced to the phrase “ROS levels in macrophages”.

Instead of assuming that ROS are omnipresent in and around the cell once produced, studies aiming to provide new mechanistic insights into direct or indirect antimicrobial functions of ROS in macrophages have to carefully consider the parameters delineated in this review, such as macrophage type, ROS-inducing stimulus, ROS sources, ROS subspecies and their specificity and selectivity for their targets, the compartmentalization of ROS production and the corresponding ROS probes, scavengers and inhibitors.

## Figures and Tables

**Figure 1 antioxidants-10-00313-f001:**
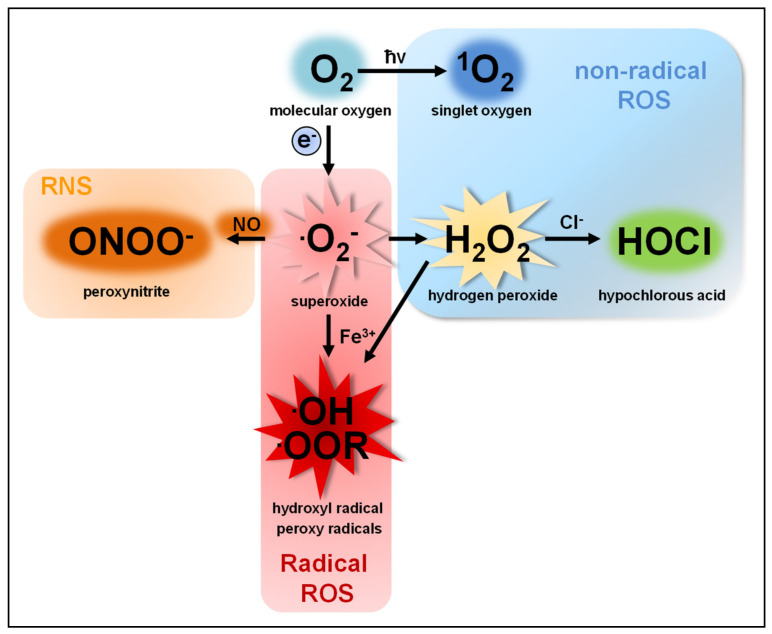
Reduction of O_2_ generates O_2_^●−^, which acts as precursor for all other ROS subspecies produced by cells. O_2_^●−^ quickly dismutates to H_2_O_2_. H_2_O_2_ can be converted enzymatically by myeloperoxidase (MPO) to OCl^−^ or by ferric iron (Fe^3+^) to ●OOR or ●OH. The oxidation of H_2_O_2_ by Fe^3+^ is called Fenton reaction [1] resulting in ●OH as reactive intermediate [2], however, this reaction rarely takes place in cells [3]. Similarly, the excitation of O_2_ to ^1^O_2_ by radiation (ħv) rarely takes place in animals [4,5]. Of note, nitric oxide (NO) and peroxynitrite (ONOO^−^) are reactive nitrogen species (RNS) and not ROS.

**Figure 2 antioxidants-10-00313-f002:**
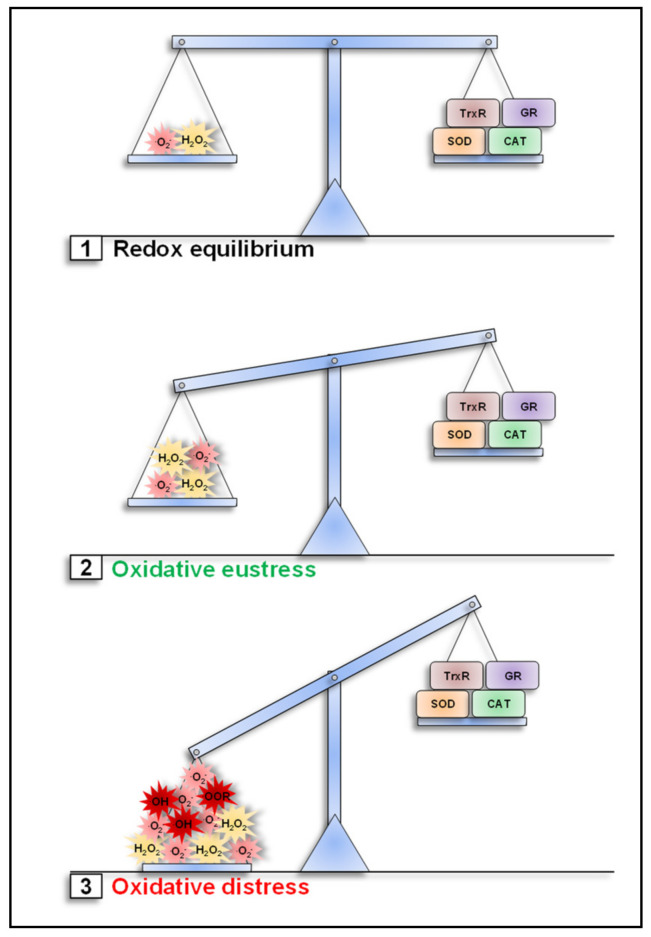
The redox status of the cell is influenced by ROS production on the oxidative side and the antioxidant defense system on the reductive side. (**1**) If ROS producing and eliminating factors are in equilibrium, redox balance is achieved, which represents the normal status in cells. (**2**) ROS production can be increased and exceed the capacity of the antioxidant defense system. If excess ROS fulfill important cellular functions, this is called oxidative eustress. (**3**) Continuous ROS production beyond the level required for cellular functions leads to oxidative damage and is called oxidative distress. Thioredoxin reductase (TrxR), glutathione reductase (GR), superoxide dismutase (SOD), catalase (CAT).

**Figure 3 antioxidants-10-00313-f003:**
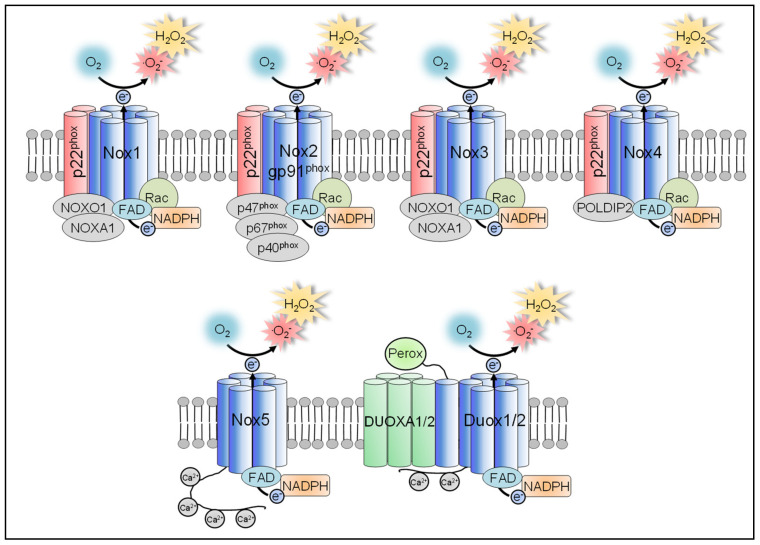
The enzyme family of NADPH oxidases consists of Nox1, Nox2, Nox3, Nox4, Nox5, and the dual oxidases Duox1 and Duox2. Different stimuli can trigger ROS production by Nox1, 2 and 3, whereas Nox4 is considered to be constitutively active. Nox5, Duox1 and Duox2 are activated by calcium binding through their cytosolic EF-hand calcium binding domains.

**Figure 4 antioxidants-10-00313-f004:**
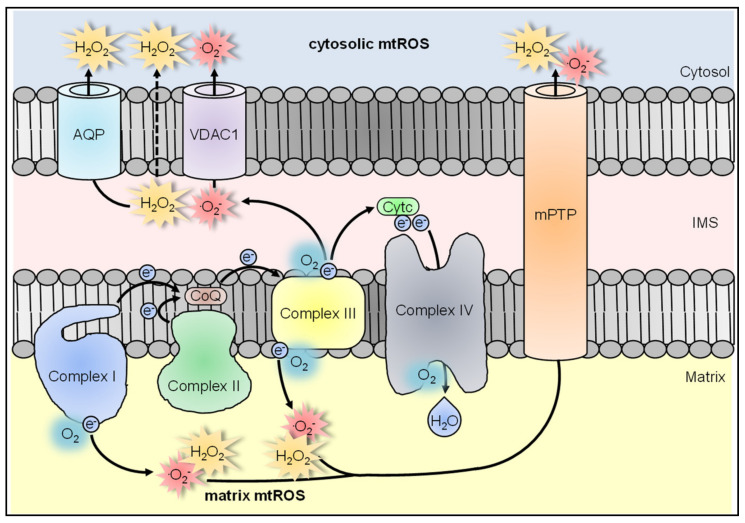
Electrons are shuttled from NADH and FADH_2_ through the four complexes I-IV of the electron transport chain (ETC) and the electron carriers coenzyme Q (CoQ) and cytochrome c (Cyt c). Mitochondria generate ROS mainly through complexes I and III. Matrix-derived O_2_^●−^ and H_2_O_2_ can only reach the cytosol after mitochondrial permeability transition pore (mPTP) opening or damage caused to mitochondrial membranes. From the intermembrane space (IMS), H_2_O_2_ can reach the cytosol through diffusion or aquaporins (AQP), whereas O_2_^●−^ can cross the outer mitochondrial membrane only through membrane channels such as voltage-dependent anion channels (VDAC).

**Figure 5 antioxidants-10-00313-f005:**
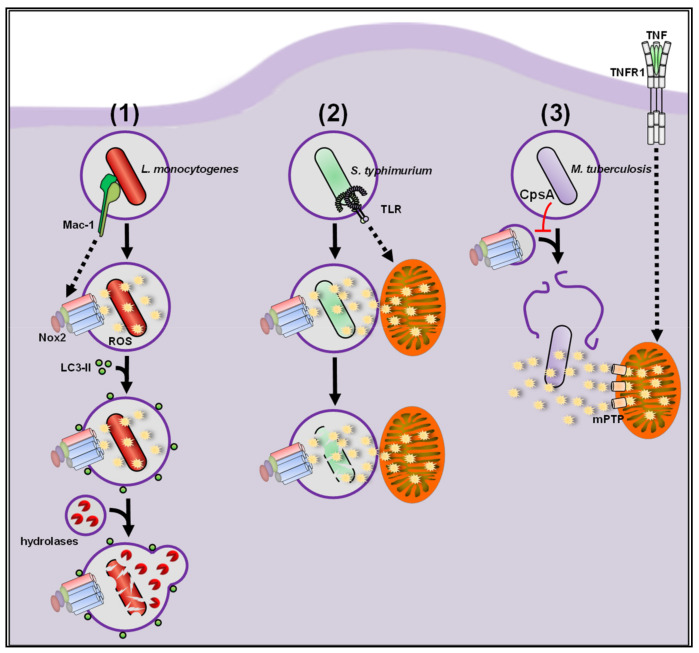
(**1**) During infection with L.m., tissue macrophages activate a highly antimicrobial phagocytic pathway called LC3-associated phagocytosis (LAP). LAP induction strictly depends on production of extracellular ROS by Nox2, which induces the eponymous recruitment of LC3 to phagosomes. These so-called LAPosomes show enhanced fusion with lysosomes, leading to improved killing of L.m. by macrophages. (**2**) S.t. can evade degradation in the phagosome by using oxidative stress for their benefit. ROS production by Nox2 contributes to S.t. killing but, on its own, is not sufficient for complete eradication of the pathogen. Mitochondria are recruited as additional ROS source via Toll-like receptor (TLR) signaling to overcome the antioxidative capacity of S.t. resulting in its degradation. (**3**) Mtb inhibits Nox2-mediated ROS production and subsequent LAP induction through its virulence factor CpsA and quickly escapes into the cytosol. Macrophages activate excessive production and release of mtROS through the mitochondrial permeability transition pore (mPTP) via TNF receptor 1 (TNFR1) signaling to combat cytosolic Mtb.

**Figure 6 antioxidants-10-00313-f006:**
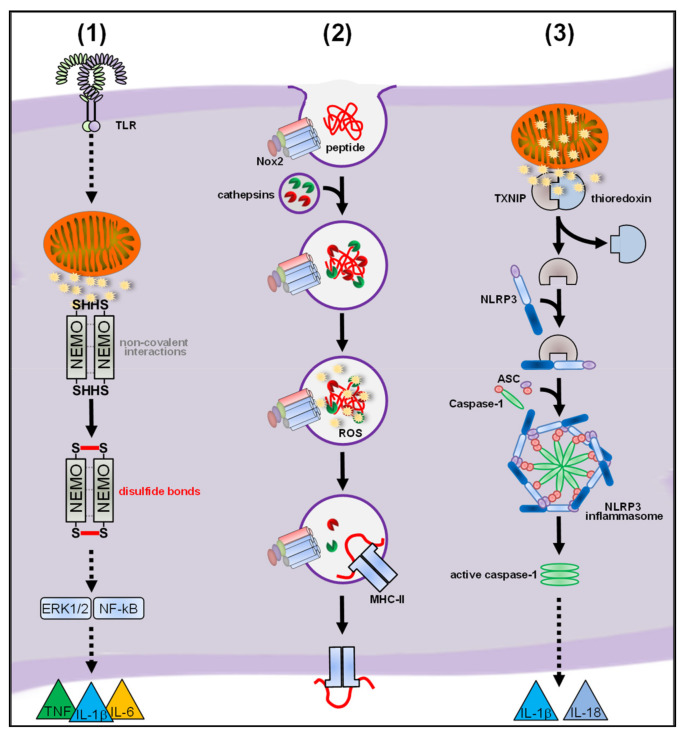
(**1**) L.m. infection induces cytosolic mtROS production by complex III of the electron transport chain (ETC) in a Toll-like receptor (TLR)-dependent manner. These cytosolic mtROS covalently link the Iκb kinase (IKK) complex subunit NF-kappa-B essential modulator (NEMO) via disulfide bonds. This covalent linkage of NEMO is crucial for proinflammatory signaling and cytokine secretion. (**2**) After phagocytosis, Nox2-mediated ROS production is induced and the phagosomal ROS inactivate the cathepsins L and S in a redox-dependent manner. This inhibits excessive proteolysis of engulfed peptides and promotes proper presentation of antigens by major histocompatibility complex (MHC)-class II molecules. (**3**) mtROS disrupt the interaction of thioredoxin-interacting protein (TXNIP) with thioredoxin, enabling its interaction with inflammasome subunit NLRP3. This step is crucial for assembly of the NRLP3 inflammasome and subsequent caspase-1 activation and cleavage of IL-1β and IL-18.

**Figure 7 antioxidants-10-00313-f007:**
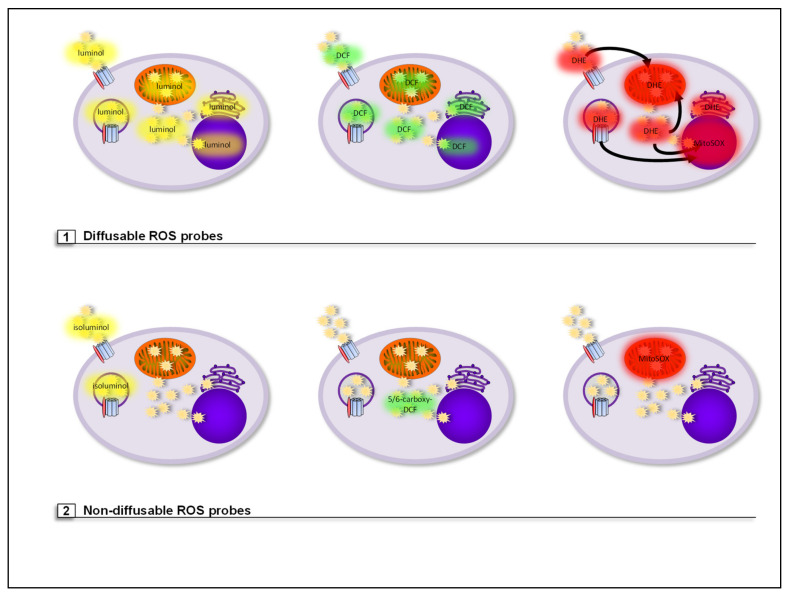
(**1**) Diffusible ROS probes are not retained or targeted to a specific cellular compartment. Luminol is a luminophore that reacts with all types of ROS. Luminol-based chemiluminescence represents a quantitative value for the amount of ROS generated but since this probe is diffusible, both extra- and intracellular ROS, i.e., total cellular ROS, are detected. H_2_DCF-DA is a cell-permeable derivative of fluorescein and one of the most commonly used ROS probes. H_2_DCF-DA is often referred to as intracellular ROS probe. However, it is diffusible and therefore does not only remain in the cytosol but also reaches cell organelles and diffuses back into the extracellular space. Therefore, like with luminol, only total cellular ROS can be detected with this probe. DHE is a red fluorescent probe that rather specifically reacts with cellular O_2_^●−^ resulting in red fluorescence. However, it easily crosses cellular membranes and therefore can be oxidized by O_2_^●−^ anywhere in and outside of the cell. Therefore, with this probe only total cellular O_2_^●−^ production can be detected. Oxidation of DHE generates two products, 2-hydroxyethidium and ethidium. Both products bind to DNA, which highly increase their fluorescence. This may lead to false interpretations regarding the localization of ROS production, namely DNA containing organelles such as the nucleus and mitochondria, when fluorescence microscopy is used to analyze ROS production. (**2**) Some ROS probes are either cell-impermeable, targeted to or retained in a specific cellular compartment. Isoluminol is a cell-impermeable derivative of luminol. This makes it ideal for exclusive measurement of extracellular ROS, which also includes ROS produced into the lumen of endosomes and phagosomes. 5/6-Carboxy-DCF is a derivative of H_2_DCF-DA that contains two additional carboxyl groups that enhance its hydrophilicity and therefore strongly increase its retention in the cytosol. Therefore, 5/6-Carboxy-DCF can be used to specifically detect cytosolic ROS levels. MitoSOX is accumulated in the mitochondrial matrix and specifically detects O_2_^●−^.

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
