# Peer review of "Functions of ROS in Macrophages and Antimicrobial Immunity"

_antioxidants, 2021, doi:10.3390/antiox10020313_

Round 1

Reviewer 1 Report

Finally a review article I have been waiting for! I wanted to write a review on a similar topic a few years back but time didn't allow it. I am glad others have thought the same and it's finally here. This is a well-written and a comprehensive review on a topic that many could relate. This review definitely needs to be published as soon as possible. I have a few comments for the authors to consider:

  1. While authors have done a god job illustration the difference between oxidative eustress and oxidative stress, I think it will significantly improve this part of the review if the authors could further elaborate the concept of oxidative eustress. Is this a widely known term? How and when do we exactly consider that the cellular oxidative state is on a eustress or an oxidative stress state? Can we consider phytochemicals that mildly increase the oxidative stress level in cells an inducer of oxidative eustress? Can we give examples of inducers of oxidative eustress? And is it related to the activation of the NRF2 or NF-kB pathway?
  2. Figure 7: I suggest changing the color of the figure to something light. In its current form, it is difficult to see the contents of the darker portions when printed in black and white.
  3. In Section 7, there are other ROS source inhibitors the authors missed to include. Example is the NOS4 inhibitor GKT137831. There are possible other ROS source inhibitors not included in the current form. Please expand search and include.
  4. There are a few minor clerical errors here and there. These can be corrected if the authors go through the entire manuscript again and carefully check grammar, spelling, choice of words that can be improved, etc.

Reviewer 2 Report

Objectives: This review gives an overview of ROS for the antimicrobial defense of macrophages. Authors point towards the relevant question of the ROS source for the biocidal action executed by macrophages. Additionally, this review completes with briefly describing the application of commonly used ROS probes/scavengers/antioxidants for analyzing involvement of ROS and their suitability for the identification of the respective ROS sources.

This excellent review addresses an important topic of the immune defense executed by macrophages, namely the role and function of ROS, and points towards the essential question where precisely these ROS are generated. Since ROS, per definitionem are short living compounds, it is clear, that a targeted action of ROS requires physical vicinity of the two reaction partners. In this context, authors are completely right in pointing towards the relevance of the precise site of ROS production instead of simply considering the overall ROS content, as is frequently done. Overall, this review is highly relevant, is well structured, written in a comprehensive way, and the figures support the text in an optimal fashion. The recommendations regarding the use and interpretation of data obtained with commonly used experimental systems for analyzing ROS is of particular interest for the readers and suitable for encouraging further research in this field.

There are only minor points, which could enhance the high quality of the manuscript:

  • Please spell out all abbreviations in the text when using them the first time (VDAC: line 40; TLR: line 299; NAC & DPI: line 336 (NAC is explained only in line 911); CSpA: line386; RIPK: line 395; MLK1: line398; MPO: line 423; MyD88 & TRAF: line 430; FGR: 453; PKC: line 522; IKK & NEMO: line 575; MEF: line 579; ASC: line 690; CPT1: line 741; Lrp3: line 755; TXNIP: line 778; noGFP2-Orp1: line 788).
  • Please spell out abbreviations in all the figure legends too.
  • In the legend to Fig. 1 funny signs appear (not recognized references?) in line 35.
  • Please subscript the stoichiometric index H2O2 in the legend to Fig. 4. in line 159.
  • The role of ROS in the pathogenic of certain viruses is very important and it would help to understand the paradoxical situation outlined in the last sentence (line 524). The reader would benefit if authors could add the information that the ROS production elicited by the is detrimental for the host, and that Nox2-deficiency abolishes the pathogenicity.
  • Not clear are the two sentences of line 568-570. Please try to rephrase.
  • In the line 969 funny format of a reference appears. Please correct.
